# The Study of Alternative Fire Commanders’ Training Program during the COVID-19 Pandemic Situation in New Taipei City, Taiwan

**DOI:** 10.3390/ijerph19116633

**Published:** 2022-05-29

**Authors:** Sheng-Chieh Lee, Ching-Yuan Lin, Ying-Ji Chuang

**Affiliations:** Department of Architecture, National Taiwan University of Science and Technology, Taipei 10607, Taiwan; linyuan@mail.ntust.edu.tw (C.-Y.L.); d9413005@gmail.com (Y.-J.C.)

**Keywords:** VR projection, incident commanders, building fire, training, COVID-19

## Abstract

In Taiwan, firefighters are responsible for transporting confirmed coronavirus cases to hospitals or epidemic prevention hotels. During the epidemic, in order to reduce the chance of cross-infection between the general public and the fire brigade, traditional collective mobilization training was limited. As a solution, this study combines a fire command course (C1) and a VR simulation for training fire station captains (VRST) and then applies the pre- and post-test evaluation approaches and the after-class questionnaire to summarize the learning effectiveness. The results show that, from a total of 244 captains that were trained, the after-training scores are significantly better when compared with the scores before training (*p* < 0.05, Cohen’s *d* > 0.8). Additionally, the post-test scores collected during the epidemic (*n* = 158) have no significant difference compared to the ones taken before the outbreak (2019, *n* = 86) in terms of sizing up, decision making, and safety management. The training results are still improving. The after-class questionnaire showed that most trainees believed that VRST helped to improve their confidence as incident commanders (ICs) (mean = 4.63, top box = 66.98%), with the novice and suburban groups exhibiting more significant improvements in command ability after the training. After the pandemic, VRST can continue as a regular training method for ICs. Because of the intervention of VRST, the differences in the command experience between different IC groups can be compensated for.

## 1. Introduction

### 1.1. Preface

In 21 January 2020, the Central Epidemic Control Center (CECC) confirmed the first case of COVID-19 in Taiwan [1]. Since then, the CECC has issued the latest data daily [2] and has also issued regulations regarding the restriction of people’s activities according to the level of the epidemic’s development [1]. In Taiwan, firefighters not only perform disaster relief work but they are also responsible for emergency rescues before getting to hospitals. Regional fire departments must transport COVID-19 patients (including those confirmed through independent inspections and regional screening stations) to centralized quarantine stations [3,4]. Therefore, during the epidemic, firefighters’ jobs are just as high-risk as those of medical staff [5]. At the same time, fire departments must abide by the rule of stopping indoors intensive training of more than five people [6]. Joint exercises between different teams and the annual disaster prevention and rescue exercises have also been suspended [7]. The purpose of the lockdown exercises and group training is to reduce social contact in order to reduce the risk [8] of firefighters infecting each other [9]. The lockdown strategy also reduces the opportunities for joint disaster relief exercises. After the lockdown is over, firefighting and disaster relief work may be affected due to fire captains’ lack of training and the practice to make up for the lost time [10]. Past studies have used virtual environments based on ship fires to train firefighters on board [11] and discuss the effectiveness of virtual reality for administering spatial navigation training to firefighters [12]. Similar VR training research has also been applied to fire extinguisher exercises for general people [13]. These results indicate that virtual reality training, if constructed and implemented properly, may provide an effective alternative to traditional training methods. In order for fire captains to continue their training under safe conditions during the epidemic [13], the use of virtual reality (VR) technology to assist in learning has become more important [14,15]. 

### 1.2. Literature Discussion

With the more contagious SARS-CoV-2 Alpha variant [16] introduced to the community, May to June 2021 was the peak period of the COVID-19 epidemic in Taiwan [17]. New Taipei City has the largest number of confirmed cases—a total of 6930, accounting for 47.5% of the local cases in Taiwan (statistics until 30 October 2021) [18]. Prior to this, the New Taipei City Government conducted a tabletop exercise [19] of the urban blockade strategic plan for COVID-19 to prepare for the blockade. Meanwhile, the CECC issued an order restricting gatherings of people, and the fire department stopped all fire drills. Unfortunately, in June 2021, in Changhua County, another city in Taiwan, a fire broke out in an epidemic prevention hotel. The fire took 9 h to put out, resulting in four deaths and twenty injuries. One of the victims was a firefighter. This case reminds us of the importance of continuous training for fire commanders during the pandemic.

On fire grounds, incident commanders (hereafter referred to as ICs) usually make decisions under mental stress and time pressure [20,21]. Coupled with the uncertainty of infectious diseases, ICs’ decision making at fire scenes is more difficult, because the actions they decide on may affect the safety of the firefighters during the rescue processes and the lives of people waiting to be rescued [22,23]. However, traditional fire and disaster relief training is a collective action [20], and firefighters from various units may increase the risk of mutual infection due to collective training. 

Taiwan’s fire service-centric emergency management system treats the fire chief of the first group arriving at the accident scene as the first-line incident mitigation commander [24]. These fire chiefs use a hierarchical structure system to implement a command-and-control style to organize response activities on-site. They are usually the lowest-leveled ICs (hereafter referred to as IC-01s) in the command chain and are therefore responsible for the initial assessment, decisions on the initial actions, and providing accurate and informative reports to higher officers and the command center [15]. They communicate with the fire brigade, other command levels, those involved in the incident, other emergency managers at the scene, bystanders, and the media. In order to reduce the casualties and property losses caused by building fires, fire brigades often conduct simulation drills to improve their ability to respond to fires. Fire drills are usually based on situational assumptions, and the commander’s response behavior is trained based on the development of the situation [25]. Traditional fire drills take place in a real, operational building environment, while the fire officer conducting the training communicates with participants through radio or writing. According to the simulated events reported by the host, the trainee IC controls the corresponding personnel and vehicles to carry out operation training in scenes that do not have real fire and smoke. For IC-01s, fire drills provide an opportunity to practice training contents outside the classroom [26]. Because the traditional fire drill is carried out through live simulation training, it is abbreviated as live simulation training (LST). LST is the traditional method during which trainee ICs will have frequent contact with firefighters and building stakeholders in the operation. During the COVID-19 pandemic, the fire brigade should reduce collective mobilization to lower the chance of cross-infection between teams.

LST is not the only way to achieve the purpose of situational drills [27,28]. Over the past 20 years, advancements in science and technology and VR research have had a profound and positive effect on the value and utility of simulation-based training [29]. VR is the use of a computer analogy to generate a virtual world of three-dimensional space, which makes the users feel as if they are in the real world and can observe things in three-dimensional space instantly without restrictions [28]. With VR-based simulation training, it is possible to achieve the benefits of safe, convenient, and planned repetitive training [30].

A literature review shows that the effectiveness of VR has been helpful in increasing surgery success rates [31,32] and improving police safety when on duty [33], and it has effectively been used in earthquake safety training [34], fire extinguisher training [13], fire and rescue training [35,36], and in increasing adherence to meditation practices [37]. Most academic research on simulations involves the use of head mounted displays (HMD), and the sound comes from headphones on both sides. They conceptualize VR as a “model training method” that allows one to train without the normal limitations of the physical world [38]. This type of training has been shown to have a positive effect on skill-based training [13]. However, these skills usually have fixed operating procedures, which are generated by users through controllers and virtual characters (or situations) in the VR, and they complete tasks according to steps or fail due to operation errors. The virtual event decision points and options in this type of game are limited, but real fire is a dynamically changing environment [39]. Fire dynamics vary over time and with intervening factors, and each fire scene is different. In addition to technical skills, firefighters training to become IC-01s need supplementary education and should focus on improving “non-technical skills” [40]. This part of the training program includes sizing up, decision making, fire communication, teamwork, and leadership (National Fire Chiefs Council, NFCC) [41]. In the process of training, experienced assessment instructors should play the role of facilitator and adjust the accident development according to the response of the trainee IC [39]. This study hypothesizes that VR Simulation Training (hereafter referred to as VRST) can provide another option for fire commander training in the post-epidemic era. Figure 1 illustrates the difference between traditional live simulation training (LST) and VR simulation training.

In order to facilitate conversation between the instructor and the students, the evaluator can clearly observe the actions of the students giving the instructions and, at the same time, consider whether the head-mounted display (HMD) may cause dizziness in the students or come in contact with infection. This research uses projection VR to conduct the experiment. There are not many studies on non-technical IC fire field command decision simulation training using projection VR. Experimental data on physical training during the COVID-19 pandemic are even more lacking. This study is the first experimental program to implement VRST in the fire department during the epidemic in Taiwan. The results will provide important experimental data for the implementation of physical courses of IC fire command training using VR during the COVID-19 pandemic. 

The purpose of the study is to explore the following three questions: (1) Is there a better performance in terms of command capability through VRST’s IC-01 training (compared to traditional training)? (2) Do participants feel that VRST helps them perform splendidly as IC-01s? (3) Will the adjustment of the epidemic prevention measures affect the training effect of VRST? The research will use experimental results to confirm the research hypothesis (VRST is an effective training method for IC-01s during the epidemic). 

The New Taipei City Fire Department (NTFD) provides fire protection and life safety services to an area of 2053 square kilometers with a population of nearly 4.02 million. The city has a diverse range of structures, from densely populated residential and high-rise office buildings to a mixture of manufacturing and industrial complexes. During the daytime, the population increases to more than 5 million in New Taipei City. The city also contains 23 universities, 55 hospitals, 3 highways passing through the city, and several railway systems, making it the most populous city in Taiwan. The fire department is composed of 73 fire stations, including 5 special search and rescue stations. There is one captain and two to three company officers working for each fire station. The area under the NTFD’s jurisdiction is divided into eight districts and twelve battalions, and each battalion has a battalion chief responsible for supervising its action. The fire department employs a total of 2500 uniformed personnel in operations, fire protection, and support services. Before the outbreak of the epidemic, New Taipei City was the only city in Taiwan to use VR in conducting fire field commander training. After the pandemic, if compliance with the epidemic prevention protocols is continued, the simulation training will provide a safe and cost-effective alternative to post-pandemic fire and rescue drills [27]. 

## 2. Materials and Methods

### 2.1. Experiment Process

The fire commanders of the NTFD were selected as the research objects. This study has collected training data before and during the outbreak. Interviews and questionnaires were used to find out how trainees respond after participating in the study. In Taiwan, during the early stages of the fire, the dispatch center assigns the closest fire brigades to the fire site. Generally, firefighters and vehicles from three fire brigades will be dispatched. If the fire expands, the battalion mayor will be assigned by the dispatch center to take over the command. The subjects of this training and testing are all fire brigade captains and agents (company officers) of the NTFD. The total sample for the study includes 244 fire and emergency rescue workers working in the fire brigade. Among them, 239 (98%) men and 5 women (2%) are included. The mean age of the total sample is 39.3 ± 6.2 (min 27, max 52) years, and the average seniority of an IC is 5.29 ± 5.15 years. Additionally, 58 (23.77%) are captains and 186 (76.23%) are company officers. There are 184 (75.4%) ICs with service locations in urban areas and 60 (24.6%) with service locations in suburban areas, and 61 (25%) subjects have less than one year of experience as an IC in the fire brigade. During the COVID-19 pandemic, 158 (64.8%) served as first-line commanders (IC-01s) at the accident scene. Since the subjects are all firefighters and had regular training, 96.7% of the subjects have exercising habits (Table 1). 

A fire brigade captain is assigned two operating shifts, so he needs to work for 2 consecutive days (48 h) and then take a day off (24 h). A complete shift roster includes 162 captains (and company officers), with a minimum staff of 122. This training takes one week, so only 80 to 90 captains (and company officers) can participate each year. In order to ensure the accuracy of the baseline measure (pre-experiment test), the captains and company officers participating in the experiment are receiving the IC training course combined with VR (VRST) for the first time. 

Every trainee who is trained (tested) in the experiment has undergone LST training before VRST. LST is continuous, while VRST is a one-time experiment. This experiment regards the pre-test scores of the C1 course as the scores of the traditional training (LST) (the control group before participating in VRST). Figure 2 shows the timeline of two training sessions. The training and testing for all the captains and company officers of the NTFD can be completed within 3 years (2019 *n* = 86; 2020 *n* = 82; 2021 *n* = 76).

This study incorporates the Fire Command Course 1 (hereafter referred to as C1) of the NTFD. A total of 12 classes were opened during April 2019 to October 2021. The pretest was completed before training to obtain the baseline data, while the post-test will be completed after the training. The scores of the group before (control group) and the group after the training (experiment group) will be compared to find the differences [42]. In order to evaluate the trainee’s ability to handle the fire site and the effectiveness of the 3-day C1 course (8 h per day), building models and various dynamic scenarios are put into the simulation. To safely and efficiently rescue the victims in the building, the trainees must act according to the scenario at hand and give clear orders to firefighters (or teams). Figure 3 and Figure 4 show the entire experimental process and the trainees’ operation line in the training ground. The experiment description and pre-test were conducted on the first day, and C1 was conducted on the mornings of the second, third, and fourth days, respectively, reviewing the basic knowledge of fire command, fire communication, and fire safety operations. Fire field virtual reality simulation training (VRST) was conducted every afternoon, and post-tests and questionnaires were conducted on the fifth day.

### 2.2. Training Setup

Jacobson [43] divided VR into four categories depending on how the visuals are produced: Desktop VR, Simulator VR, Projection VR, and Immersion VR. Incident commanders (ICs) use radios in fire scenes to issue instructions [44]. We believe that trainees will have the most realistic experience if they use the same tools (radios) in simulation training [45]. To provide trainees with visually and audibly dynamic information, this experimental field uses Projection VR as the main visual image presentation, functional compartments, expanded three-projection screens (each with a length of 4.8 m a height of 2.6 m in a space of 485 m^2^), and audio equipment. The computer simulation software uses the Advanced Disaster Management Simulator (ADMS) system. The system can create 3D simulation scenes (i.e., flames, a hazard location, weather conditions, specific effects, the number of victims and their locations, etc.) in advance. The simulation process can be based on the time records of real cases. When the exercise begins, the trainee in the role of the IC can use a wireless radio to give instructions. The instructor (facilitator) of the rear center console will command the firefighters in the VR to perform actions according to what is heard on the radio. The trainee can use the joystick to move the character and also its field of view. The character is an IC in the fire scene, and it can be made to move around freely and make observations in order to find key points. The real movements of the user are translated into movements within the virtual space using a joystick and wireless radio [30]. An elevated view of the actual operation and evaluators’ position is shown in Figure 5a. In addition to the main training ground, the simulated training ground is surrounded by a grouping operation area consisting of 10 operation units, each unit having a space size of 3 m × 3 m, and the configuration equipment is shown in Figure 5b.

### 2.3. Virtual Scenario Design

According to Heldal [46], one way to make training scenarios more realistic and reasonable is to import actual recorded data from previous rescue operations into the data of the simulated scenarios. These scenes are not only realistic but also based on actual accidents. Cha [30] notes that fire training using VR can provide a considerable amount of second-hand experience for the general public, firefighters, and ICs, allowing them to make appropriate and organized decisions when encountering actual fire incidents, thus improving people’s safety. Therefore, this study uses five-story apartments, which are involved in the most fire accidents in Taiwan [47], and tin houses, which often cause firefighter casualties, as the simulation cases, as shown in Figure 6. These simulated situations use on-site noise playback and evaluators as role players (as in Figure 5a) to provide trainees with information and improve their sense of presence and psychological pressure. In the pre- and post-test scenarios, the accident scenario will have five stages: (1) receiving the task, (2) en route, (3) arriving at the scene to perform rescue operations (action plan), (4) emergency handling, and (5) transferring of command. Each trainee is required to complete the fire command operation in the above situation within 15 min, and the unfinished actions are not included in the scoring, as shown in Figure 7. The time constraints during the test provided the trainees with psychological stress similar to that of a fire scene emergency [48]. 

### 2.4. Instrument Design

IC training needs to be adjusted mid-activity in order to focus on correct decision making and to educate current and future ICs to adopt appropriate accident mitigation techniques. Computer simulation is used to create virtual environments to enhance trainees’ knowledge and experience so as to reduce hazards at on-site fire grounds or operational training activities [30,49]. Historical fire cases are reconstructed using VR during the courses to allow trainees to implement simulation training immediately after learning the knowledge of the courses (Figure 8). The role in the VRST course is shown. During the training, the lecturer acts as the facilitator. During the test, two evaluators (three, four) will be added. They also play the roles of those with relations to the accident scene, the support unit, or the reporter (the role of the other facilitators). The evaluators can observe the learning performance of the students through the items in the evaluation form. Researchers [35,50] also used the same method to evaluate the IC’s computer simulation training.

Since this research needs to evaluate the performance of fire officers who are fire station captains or company officers as ICs, five experts were invited to review it after referring to the content design of the evaluation form above. The experts include two PhDs in fire protection and three ERCA core instructors. The evaluators during the experiment were three squadron chiefs with more than 10 years of experience as fire field commanders. The ability indicators on the basis of which trainees are evaluated are established with reference to the NFPA 1021 Standard for Fire Officer Professional Qualification (Fire Officer Ⅰ and Ⅱ). Among the important indicators are the ability to analyze the situation of the fire, the ability to use the content of the rescue plan, the ability to allocate resources, the ability to communicate with radio and verbal commands, the ability to allocate and track task assignment personnel, the ability to communicate and coordinate, the ability to implement safety management, the ability to respond to emergencies, and the ability to transfer the command. According to expert opinions, the nine types of abilities are organized into three dimensions so as to correspond to the contents of C1 3 courses. The evaluation form has a total of 40 checked items, and the total score of each dimension is 100 points. We tested 12 small samples with the form in advance. The calculated α value was 0.8. The design of the evaluation form has obtained sufficient reliability and validity. Therefore, this experiment is based on the evaluator’s record of the degree of practice of the trainees in the implementation of the project in the experiment as a quantitative basis. The pre-test and post-test evaluation indicators of this experiment are the same, as shown in Table 2.

The contents of the trainees’ subjective feedback questionnaire referred to [50], after Chinese and English native speakers confirmed that the meanings are correct. The questionnaire is produced using the five-point Likert Scale. The choices are strongly agree, agree, neutral, disagree, and strongly disagree. The questionnaire has eight questions which ask about perceived usefulness [51] and learning satisfaction. 

After 2020, LST was interrupted or stopped due to the epidemic’s impact (as shown in Figure 2). In order to improve the experiment’s reliability, a total of 158 trainees participated in the training after 2020. The trainees’ self-assessment was conducted once before and once after the training, respectively (two times in total). The trainees’ self-assessment form is composed of the nine ability indicators in Table 2, using the Likert Scale (full score is 5 points), and they subjectively evaluated their ability performance.

### 2.5. Data Analysis

In this study, paired sample *t* tests were used to analyze the difference between traditional training (LST) and VR training (VRST) and the difference between the trainees’ self-assessment through the VRST pre-test and post-test scores. Independent sample *t* tests were used to analyze the difference between the post-test scores of the trainees following VRST before the outbreak and during the epidemic prevention period. Analysis of variance (ANOVA) was also used to determine the impact of the epidemic prevention measures taken during the epidemic on the training effect.

The post-class feedback questionnaire part used narrative statistics to illustrate different aspects of learning satisfaction. Analyses were performed using SPSS 22 statistical software. Continuous variables are reported as the mean ± standard deviation (SD), while categorical variables are expressed as percentages. All of the tests were two-tailed, and *p*-values < 0.05 were considered statistically significant.

## 3. Results

### 3.1. The Descriptive Statistics of Decision Making, Fire Communication, and Incident Safety Management 

For the pre-test and post-test objective scoring analysis part, a “pre-test” is conducted for the trainees before participating in the training, and they will move on to a “post-test” after the training is complete. The correlation between the three evaluators is determined by the Spearman correlation coefficient, and the (Cohen’s kappa) correlation between the pre-test and the post-test was 0.92 and 0.97 (*p* < 0.001), indicating that the evaluators were in good agreement. This study conducted 244 pre-test and 244 post-tests assessments (from 2019 to 2021). The statistical analysis of these data includes: descriptive statistics, correlation tests, paired sample *t* tests, and analysis of variance. This was done in order to compare the differences between the trainees before and after the overall training in three years. 

The average pre-test score of the sizing up and decision-making ability is 46.33 ± 15.72 points (full score: 100 points); the average post-test score of this ability is 74.58 ± 14.57 points. The average pre-test score of the fire communication ability is 59.50 ± 16.4 points, while the average post-test score is 75.39 ± 13.56 points. The average pre-test score of the incident safety management ability is 45.23 ± 20.87 points, and the average post-test score of this ability is 76.14 ± 16.43 points (Table 3.)

### 3.2. Compare the Post-Test Scores of the Trainees after C1 Training before the Outbreak and during the Epidemic Prevention Period

A total of four classes were trained in 2019, and a total of 86 trainees were trained. In January 2020, when the COVID-19 epidemic began to spread, in order to continue the training, this study was adjusted according to the CDC regulations [52]. There are three differences (shown in Figure 9) between the training processes during and before the COVID-19 outbreak: (1) The number of people in the training ground is limited to fewer than five people; (2) The trainees and evaluators must wear masks; (3) The trainees need to measure their forehead temperature before entering the training ground. If the forehead temperature is higher than 37.5 °C, entry is prohibited. The trainees are then tracked to see whether they have symptoms 14 days after training. If the temperature exceeds 37.5 °C, the trainee’s fire brigade will notify the training unit immediately.Besides the above differences, the rest of the training materials, VR scenarios, and evaluators remain unchanged. Four training classes were still completed in 2020 and 2021. A total of 82 trainees completed training in 2020, and 76 trainees completed training in 2021. A total of 158 trainees completed training during the epidemic.In order to understand whether or not the adjustments made in response to the epidemic prevention regulations have changed the training results, this study conducted post-tests on trainees who completed the training before and during the outbreak, and independent samples of *T* Test analyses were conducted on their scores, as shown in Table 4.The post-test scores of the trainees before the outbreak were divided into three dimensions corresponding to the three dimensions of the post-test scores of the trainees during the epidemic, and a box-shaped chart was made, as shown in Figure 10. The median of the sizing up and decision-making ability before the epidemic was 77.5, and the interquartile range (IQR) was 20, which was similar to the sizing up and decision-making ability of the training during the epidemic (median = 78, IQR = 19.25). The median for fire communication is 82.5, while the IQR is 19. Compared to during the epidemic, the median is higher (74), while the IQR is lower (22). The median of incident safety management capabilities before the epidemic was 76, and the IQR was 30, which are slightly lower than those during the epidemic (Median = 80, IQR = 20.12).

### 3.3. Explore the Correlation between Students’ Personal Background, Gender, Service Experience, and Other Data Compared with VRST Performance

By using Table 5 to explore the differences in the objective scoring results of the sample feature grouping, there are findings through the independent sample *t* tests of different groups: the novice group (less than one year of IC experience) is not as good as the group with more than one year of IC experience in the pre-test and post-test in terms of fire communication performance, but the growth rate (T2 − T1) of the novice group is better than that of the senior IC group and reaches a statistically significant difference. The suburban and the female groups also had a better learning growth rate than the urban and male groups in terms of sizing up and decision-making, and the *T* test between the two groups also showed a statistically significant difference. In terms of position grouping, there is no statistically significant difference between the captains and the company officers. 

During COVID-19, the post-test performance of the trainees in the training group was still significantly better than the pre-test performance, but the post-test scores and the growth rate in terms of fire communication were lower than those of the group that participated in the training before the COVID-19 outbreak. In addition, the average age of the novice IC group (with less than one year of IC experience) was 33 years, and the average age of the IC group with more than one year of IC experience was 41. There was a significant age difference between the two groups. To understand whether the progress of the trainees’ C1 training before and after the outbreak of the epidemic has an impact, the data of the *T* test analysis are shown in Table 6, showing the differences in the performances of the three ability dimensions. We further conducted Analysis of Variance (ANOVA) on the training data before and during the outbreak. The results showed (in Table 7) that the factors before and after the epidemic were significantly correlated with the progress of sizing up, decision making, and fire communication, but no significant differences with regard to incident safety management capabilities were found.

### 3.4. Analysis of the Subjective Feedback Questionnaire of the Trainees

A total of 158 trainees participated in the training after 2020. The monthly regular training (LST) of IC-01s was modified in consideration of the influence factors of the epidemic. In order to improve the reliability of the experiment, before and after the C1 training, the students’ self-assessment was conducted once (two times in total). After deducting the wrong samples with the incomplete data, the number of valid samples for the pre-test self-assessment is 129 (*n* = 129), and the number of valid samples for the post-test self-assessment is 115 (*n* = 115). Paired sample *t*-test analysis was used, and the results are shown in Table 8.

The results of the students’ subjective assessments were consistent with those of the objective assessments of the scoring instructors, and the scores of each ability were statistically and significantly different. The top three indicators with a high degree of effect (Choen’s *d*) were: (1) The ability to analyze the situation of the fire, (2) the ability to use the content of the rescue plan, and (3) the ability to allocate and track the task assignment personnel.

In order to understand the subjective feelings and satisfaction of the trainees for the C1 course of the VR training, an electronic questionnaire was conducted after the training. This questionnaire is divided into three constructions according to the content correlation, and the eight questions of the questionnaire have high internal consistency (Cronbach’s alpha = 0.738). The scale of the questionnaire uses the five-point Likert Scale, ranging from 1, strongly disagree (bottom box), and then increasing to 5, strongly agree (top box), and the reliability and validity have reached a sufficient level. This study collected the number of trainees participating in the C1 course during the epidemic prevention period (2020 to 2021) in Taiwan (*n* Total = 158). After consistency and reverse tests, invalid questionnaires (*n* Rejected = 52) were excluded, leaving only the valid questionnaires (*n* Total Valid = 106). The results show that the overall satisfaction average (mean) is 4.4 (out of 5 points). The trainees have good satisfaction with the C1 course combined with the VR training, and the courses and instructors have the highest degree of satisfaction (mean = 4.64, mode = 5). Most of the trainees believe that the training can help improve their confidence as ICs (mean = 4.63, top box = 66.98%). See Table 9 for details.

## 4. Discussion

During the COVID-19 pandemic outbreak, many businesses have done their best to educate employees regarding the potential symptoms of COVID-19 and enforce strict screening protocols to avoid the spreading of COVID-19 [53]. Avoiding cross-infection among firefighters is also the key to maintaining the continuous operation of the fire department. This study proposes the use of VR training to replace battalion-level fire and rescue drills during the epidemic to reduce the possibility of fire brigade contact with each other. The experimental results obtained the following conclusions:

Based on the literature review and expert discussion, this study divides the fire field command ability into three dimensions: (1) sizing up and decision making ability; (2) fire communication ability, and (3) incident safety management ability. A total of 40 execution projects were used as the basis for the measurement. In the experiment, the trainees were given a “pre-test” before the training. Since all the test subjects received traditional training (LST) before participating in the training and completed the education and testing of VR hardware before the test, the test subjects stated that they had no operational problems before starting the implementation. We can interpret the “pre-test” scores obtained by the trainees after the traditional training as the control group and the scores obtained by measuring trainees’ performances after passing the C1 course training as the experimental group.

C1 consists of a course plus VR simulation training. Two kinds of building fire scenarios were simulated through the ADMS system, and the trainees’ tests were provided in turn. The test was conducted by three senior battalion chiefs serving as evaluators. According to the performance of the trainees, they registered the degree of implementation with the evaluation form (as in Appendix A). Most of the trainees’ ability performance after the training was significantly better than that before the training in terms of all of the indicators of the *T* test (*p* < 0.05). The Cohen’s *d* values of the three dimension sizes were all greater than 0.8. It is obvious that, after the C1 course, the commanding ability of the trainees improved significantly. A possible reason for this is that Taiwanese fire commanders have not received VR training combined with historical cases, and the C1 course provides immersive, repeated exercises to achieve greater results. 

The study compared the training effects before and during the outbreak and found that the improvement in the fire communication ability during the outbreak was reduced (*p* < 0.05, Cohen’s *d* = 0.54). We further interviewed the experiment’s three evaluators. Evaluator A said: “I have observed that some trainees during the training session found themselves inattentive in the past (pre-test), so in the post-training test, they are more attentive to get those parts done well (paying more attention to area division and task grouping), but at the same time, they forgot to give an order to attempt a rescue from outside the building”. 

Evaluator B said: “The trainees who participated in the training before the outbreak had more time to rest after completing the 8 h course. There were more emergency cases during the epidemic. After the course, the fire brigade on call often had to support emergency confirmed cases. This may put pressure on the trainees’ physical and psychological conditions and affect learning performance”. 

Evaluator C said: “During the epidemic, the number of people in the room was limited, and the trainees were separated for training and could not observe each other. I think this reduced the opportunity for mutual learning. In addition, most trainees had a good performance in terms of the fire field communication skills during the pre-training test, so there will be less progress than that for the other two abilities”.

A possible reason for this is that the dispatch center of the fire station has audited the fire commander’s radio reporting discipline, so senior commanders have better radio reporting ability; thus, the difference in the ability before and after the training is small. In addition, during the epidemic, the rescue time of the fire brigade increased. According to the New Taipei City Health Bureau, in 2021, the number of suspected COVID-19 patients assisted by the fire brigade’s ambulance personnel was 7336 [54]. Past studies [55,56,57] pointed out that, during the COVID-19 pandemic, EMTs and medical staff wore PPE material and protective clothing for a long time, which caused them the physiological stress of heat strain. Some medical staff and ambulance personnel suffered from PTSD after facing a large number of patient infections and casualties. Taiwanese scholars [5] investigated the pressure reports of medical and pre-hospital rescue personnel (EMT) in Taiwan in 2021, who faced a large number of patients with COVID-19. The pressure of the rescue personnel in New Taipei City (EMT is served by the firefighters) is greater than that of Taichung City. They found that EMTs generally have more moderate burnout. These pressures indirectly affect their job performance. 

During the experiment, the government took necessary measures in response to the COVID-19 epidemic prevention regulations. During the epidemic prevention period, the training institution limited the number of people in the training ground to fewer than five and required trainees and instructors to wear masks while training. After 2 years of epidemic prevention, the course continued to maintain normal operation, and no trainees were infected with COVID-19 or caused transmission due to participation. 

After completing 12 C1 trainings, we interviewed the C1 course instructor A and B of this training (they have 20 years of firefighting experience and more than 10 years of educational work experience). They gave their thoughts on the introduction of VRST’s C1 course relative to traditional training (LST): 

Instructor A said: “The C1 course combined with VRST provides a successful example for us to train IC-01s during the epidemic prevention period. VR allows our training to review historical disaster cases, and it is easier for the trainees to understand the situation within the simulation. I have seen trainees with more positive motivation to learn during the training process because each person’s response to the situation can be recorded during the simulation training”.

Instructor B said: “I don’t think traditional LST can achieve dynamic disaster scenarios like VRST. For example, a collapsing scenario can be injected immediately during the firefighting operation in a tin house fire scene under VRST; additionally, the traditional LST needs to mobilize a lot of manpower, and the cost of each training is very high. The LST needs to borrow venues, which cannot provide frequent exercises for some operating venues, and the procedure is quite troublesome. Thus, the VRST can relieve the inaccessibility of special occupancies. I think VRST does provide the advantages of being safe and more effective, having a lower cost, saving manpower, and reducing fuel consumption. In the future, the routine training of IC-01s should be adjusted to VRST-based and LST-assisted”.

Overall, our findings showed that all of the dimensions of the post-training test scores were significantly better than those of the pre-test scores (*p* < 0.05, and Cohen’s *d* > 0.8). The overall average trainee satisfaction survey score was 4.4 out of 5, showing that the majority of trainees agreed with using VR in fire commander training models. The interview results are similar to the document discussion [34,39,43,51], and the simulation provides a low-cost, safe, and effective method. The statistical results also found that the novice and suburban groups’ trainees made more significant progress in their command abilities after training. Therefore, after the pandemic ends, VRST should become the regular training method for IC-01s. The difference in the command experience can be made up among the different IC groups by the intervention of VRST. The study results provide a relatively safe alternative training method for firefighters during the epidemic prevention period. Since most of the past studies involved the use of head-mounted displays (HMDs) as experimental tools, this study also verified the use of non-wearable projection VR during the epidemic to show the effectiveness of fire commander training.

### 4.1. Limitations

The researchers identified several limitations of the training during the interview: (1) During the test, each trainee has little time to operate. Due to time constraints, trainees do not have enough time to discuss feasible solutions in the situation, which may prevent trainees from make better decisions. (2) The 3-day after-class time does not give trainees enough time to rest, which may affect the effects of learning. (3) During the epidemic, the limited number of people in the training ground has reduced the opportunities for trainees to observe each other. These limitations can be improved in the future through curriculum adjustment and hardware equipment expansion. Additionally, VR can achieve immersion, interaction, and imagination [58], but it cannot cause physical fatigue. VRST’s learning limitation is that it cannot impose physical loads on the IC. The subjects of this experiment are professional firefighters, and the evaluation form’s content has taken into account the local fire service regulations, as the readers should be properly adjusted with the training subjects.

### 4.2. Future Directions

In addition to simulating fire rescue, VR can also create simulations of natural disasters, terrorist attacks, and a large number of casualties. In the future, researchers can discuss the application of VRST in natural disasters or rescue operations with multiple casualties and the training for response workers. VRST also provides opportunities for student feedback, practice, and repetition [59]. In the future, researchers can observe student performance through situational models. It is also possible to explore the difference between the VR situation and the real situation faced by IC students through the photographic records worn by them at the real fire scene.

## 5. Conclusions

The results of this study provide a feasible training method (VRST) for fire commanders during the epidemic prevention period to reduce the chance of contact between fire brigades. The experiment verifies the effectiveness of implementing VR in fire commander training through the objective records of evaluators and the subjective feedback of trainees. The research process also brought up several limitations of this training and suggestions which can be utilized as a reference in training units or fire departments.

## Figures and Tables

**Figure 1 ijerph-19-06633-f001:**
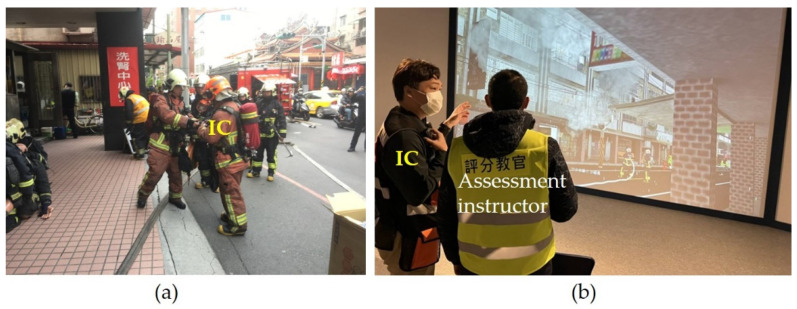
(**a**) For traditional training (LST), one IC trainee responds to the situation of live simulation training, assigning tasks to support units (one company officer and three firefighters). The scene is an operating building (not a training ground). There is no identifiable fire or smoke, and the simulated situation is only expressed through oral language. (**b**) In this study (VRST), one IC trainee responded to the VR situation and informed the assessment instructor about the task assignment.

**Figure 2 ijerph-19-06633-f002:**
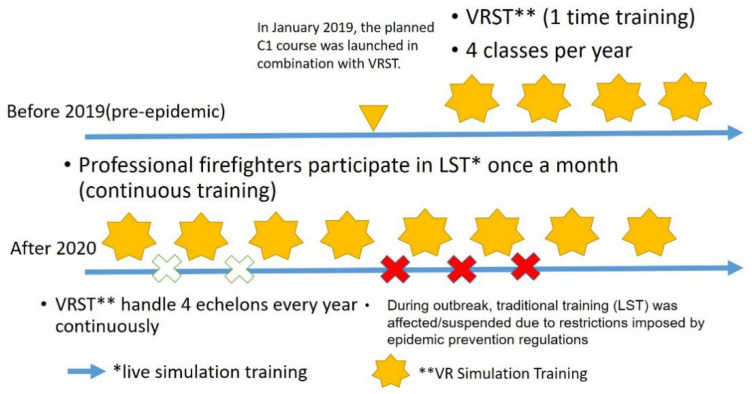
Timeline of the implementation of LST and VRST by the New Taipei City Fire Department.

**Figure 3 ijerph-19-06633-f003:**
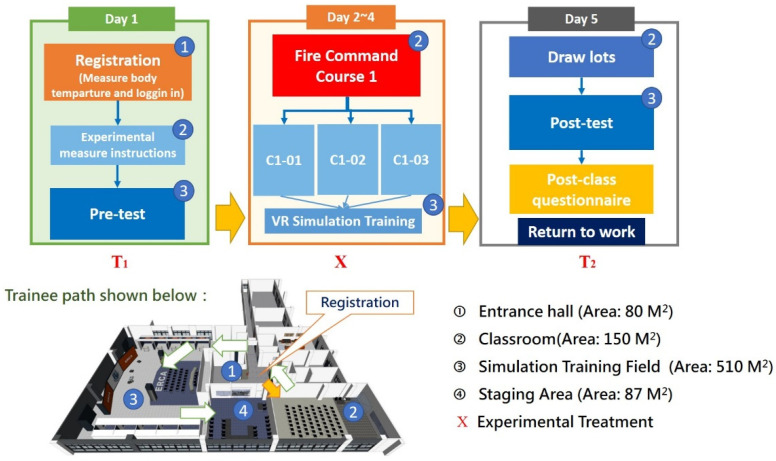
Experimental design mode and students’ movement sequence.

**Figure 4 ijerph-19-06633-f004:**
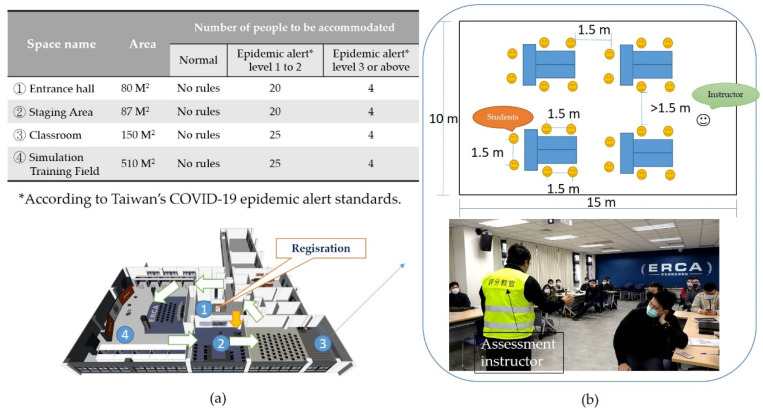
(**a**) Adjust the students’ movement sequence (1 > 2 > 3 > 4) according to the epidemic control measures and limit the number of people in each classroom according to the development stage of the epidemic. (**b**) For teaching classrooms, students are required to sit 1.5 m apart and wear masks during epidemic alert levels 1 to 2. C1 courses above alert level 3 will be changed to online courses, but VRST will be conducted normally.

**Figure 5 ijerph-19-06633-f005:**
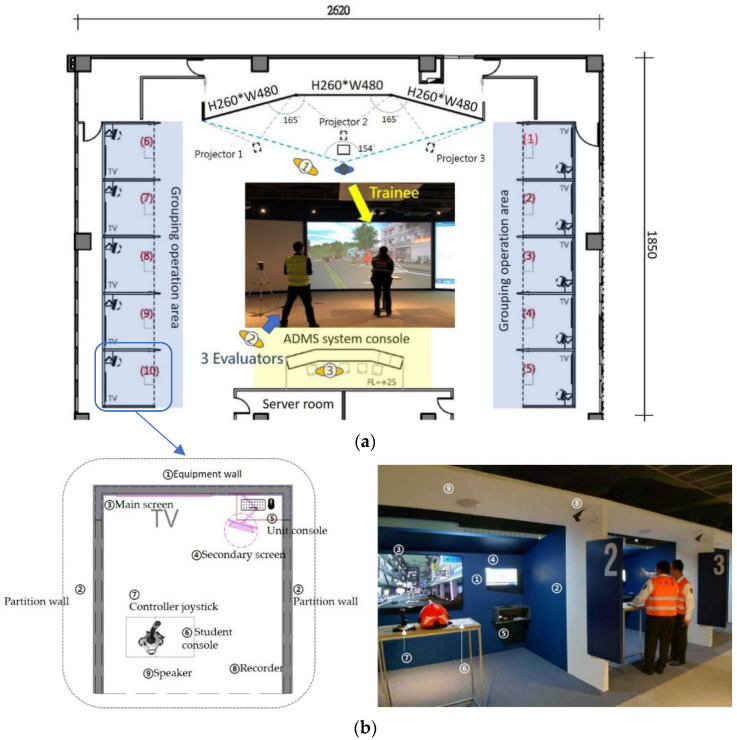
(**a**) The plan configuration diagram of the on-site simulation training field. (**b**) Equipment configuration diagram of the individual operation area. (The numbers in the sub-figures correspond to the text descriptions in the left figure.).

**Figure 6 ijerph-19-06633-f006:**
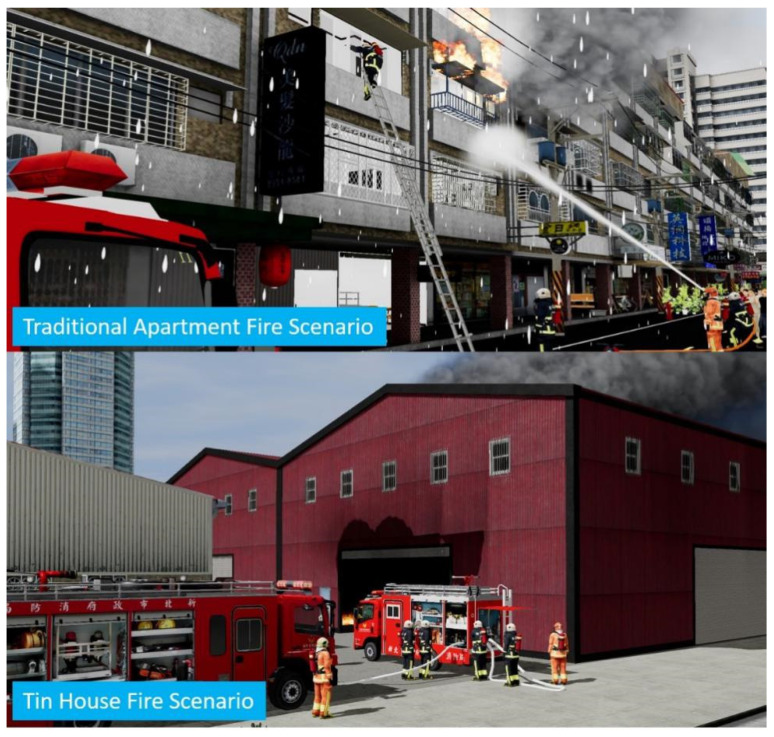
The two different types of buildings in this experiment’s fire simulation.

**Figure 7 ijerph-19-06633-f007:**
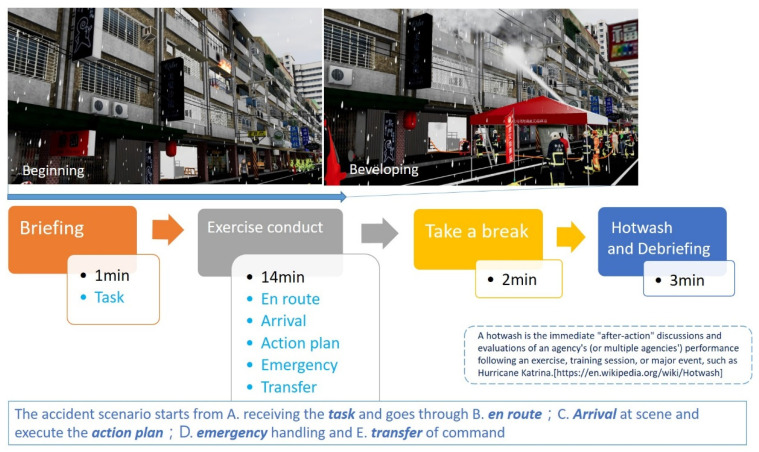
The experiment situation flow chart.

**Figure 8 ijerph-19-06633-f008:**
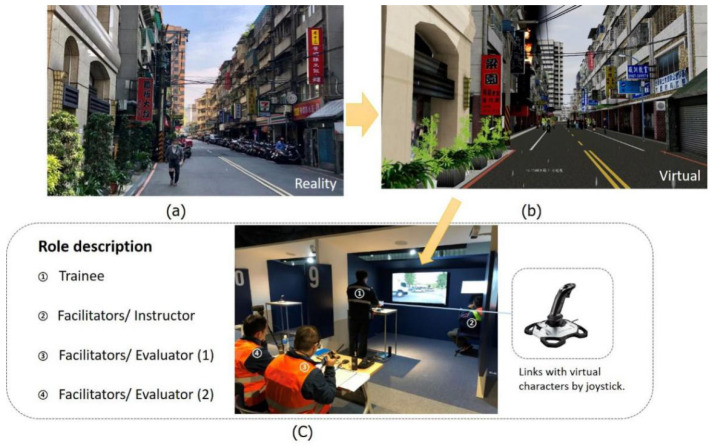
(**a**) Realistic building and street. (**b**) Historical case simulations in VR. (**c**) Interactive locations of VRST trainees, facilitators, instructors, and evaluators (operation unit used as example).

**Figure 9 ijerph-19-06633-f009:**
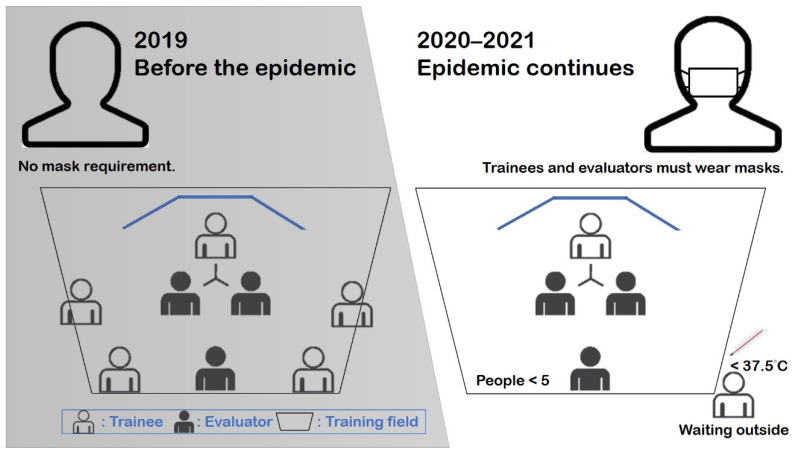
Adjustment due to COVID-19 prevention.

**Figure 10 ijerph-19-06633-f010:**
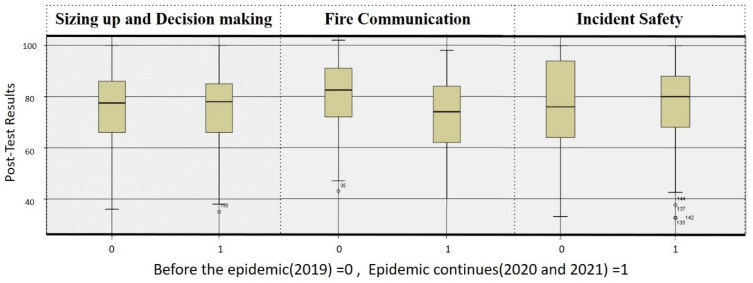
Box charts of trainees’ post-test scores before and during the epidemic.

**Table 1 ijerph-19-06633-t001:** Distributions of trainees’ characteristics and composition ratios (*n* = 244).

Trait	Ratio (%)	Mean ± SD
Age		39.3 ± 6.2
Experience as IC (year)		5.29 ± 5.15
Gender		
Male	239 (98%)	
Female	5 (2%)	
Position		
Captains	58 (23.77%)	
Company officers	186 (76.23%)	
Service area properties		
Urban	184 (75.4%)	
Suburban	60 (24.6%)	
Experience as IC		
Less than 1 year	61 (25%)	
More than 1 year	183 (75%)	
Exercising habit		
More than three times per week	76 (31.1%)	
More than once per week	160 (65.6%)	
None	8 (3.3%)	
Experienced COVID-19		
None	86 (35.2%)	
Front line commander	158 (64.8%)	

**Table 2 ijerph-19-06633-t002:** Evaluation Index to Objectively Evaluate Trainees’ Fire Command Ability.

No.	Dimension	Ability Indicators	Trainee’s Execution Projects *
A	Sizing up and Decision-making	1. The ability to analyze the situation of the fire.	A-01; A-02; A-03; A-04; A-05;
			A-06; A-07; A-08; A-09; A-10
		2. The ability to use the content of the rescue plan.	A-11; A-17; A-18; A-19; A-20
		3. The ability to allocate resources.	A-12; A-13; A-14; A-15; A-16
B	Fire Communication	1. The ability to communicate with radio and verbal commands.	B-05; B-06; B-08
		2. The ability to allocate and track task assignment personnel.	B-01; B-02; B-03; B-04
		3. The ability to communicate and coordinate.	B-07; B-09; B-10
C	Incident Safety management	1. The ability to implement safety management.	C-01; C-07; C-09
		2. The ability to respond to emergencies.	C-02; C-03; C-04; C-05
		3. The ability to transfer command.	C-06; C-08; C-10

* Number in the field corresponds to Appendix A: Dimension number and evaluation contents of this study’s evaluation form.

**Table 3 ijerph-19-06633-t003:** *T* Test Analysis of the Pre- and Post-Test Results.

Dimensions	N	Mean (SD)		M.D.	*t*	*p* ^1^	*d*
		Pre-Test	Post-Test				
Sizing Up and Decision Making	244	46.33 (15.72)	74.58 (14.57)	38.25	28.3	<0.001 **	1.864
Fire Communication	244	59.50 (16.4)	75.39 (13.56)	15.89	13.3	<0.001 **	1.055
Incident Safety Management	244	45.23 (20.87)	76.14 (16.43)	30.92	23.71	<0.001 **	1.646

^1^ ** *p* < 0.001

**Table 4 ijerph-19-06633-t004:** *T* Test analysis of trainees’ post-test scores before and during the outbreak.

Dimensions	Post-Test Mean (SD)		*t*	*p* ^1^	*d*
	2019 (*n* = 86)	2020~2021 (*n* = 158)			
Sizing Up and Decision Making	74.92 (14.96)	74.39 (14.40)	0.269	0.788	0.0361
Fire Communication	80.03 (13.35)	72.85 (13.03)	4.08	<0.001 **	0.5443
Incident Safety Management	75.93 (18.46)	76.26 (15.28)	−0.149	0.882	0.0194

^1^ ** *p* < 0.001

**Table 5 ijerph-19-06633-t005:** The objective evaluation score (mean ± SD) was obtained by the total sample (*n* = 244) passing the pre-experiment/post-experiment test and then divided into groups according to the trainee’s professional title, service attributes, and personal characteristics to perform the independent sample *T* test between each group.

Trait	N (%)	Sizing Up and Decision Making (Mean ± SD)						Fire Communication (Mean ± SD)						Incident Safety Management (Mean ± SD)					
		Pre-Test (T1)	*p*	Post-Test (T2)	*p*	M.D. (T2 − T1)	*p*	Pre-Test (T1)	*p*	Post-Test (T2)	*p*	M.D. (T2 − T1)	*p*	Pre-Test (T1)	*p*	Post-Test (T2)	*p*	M.D. (T2 − T1)	*p*
Total sample	244 (100)																		
Male	239 (98)	46.6 ± 15.7	0.052	74.5 ± 14.6	0.731	27.9 ± 15.5	0.022 *	59.7 ± 16.3	0.156	75.4 ± 13.5	0.892	15.7 ± 18.7	0.18	45.4 ± 20.9	0.303	76.0 ± 16.5	0.438	30.6 ± 20.3	0.092
Female	5 (2)	32.8 ± 10.3		76.8 ± 12.1		44.0 ± 15.7		49.2 ± 21.4		76.2 ± 16.3		27.0 ± 16.6		35.7 ± 16.6		81.8 ± 12.8		46.10 ± 21.6	
Positions																			
Captains	58 (23.8)	46.2 ± 13.7	0.969	75.7 ± 12.9	0.473	29.5 ± 14.7	0.48	60.2 ± 17.9	0.694	75.9 ± 12.8	0.735	15.7 ± 18.3	0.921	46.3 ± 17.6	0.623	78.8 ± 14.1	0.125	32.6 ± 17.2	0.483
Company officers	186 (76.2)	46.4 ± 16.3		74.2 ± 15.0		27.8 ± 15.9		59.3 ± 16.0		75.2 ± 13.8		16.0 ± 18.8		44.9 ± 21.8		75.3 ± 17.0		30.4 ± 21.3	
Experience as IC																			
More than 1y	183 (75)	43.8 ± 15.9	0.00 **	71.4 ± 14.9	0.00 **	27.6 ± 15.9	0.274	63.7 ± 15.3	0.00 **	77.6 ± 13.6	0.00 **	13.9 ± 18.7	0.004 *	42.4 ± 22.4	0.00 **	74.2 ± 17.5	0.00 **	31.8 ± 21.4	0.252
Less than 1y	61 (25)	53.9 ± 12.6		84.0 ± 8.3		30.1 ± 14.6		46.9 ± 12.9		68.7 ± 11.1		21.8 ± 17.4		53.7 ± 11.8		82.0 ± 11.0		28.3 ± 16.6	
Service area properties																			
Urban	184 (75.4)	47.7 ± 15.4	0.021 *	74.4 ± 14.3	0.773	26.8 ± 15.6	0.01 *	59.6 ± 16.5	0.842	75.8 ± 13.6	0.456	16.1 ± 18.6	0.738	46.3 ± 20.1	0.148	77.1 ± 16.2	0.123	30.8 ± 20.1	0.831
Suburban	60 (24.7)	42.3 ± 16.2		75.1 ± 15.4		32.7 ± 14.8		59.1 ± 16.1		74.3 ± 13.5		15.2 ± 18.9		41.8 ± 22.9		73.3 ± 17.0		31.4 ± 21.2	
During COVID-19																			
Front line commander	158 (64.8)	47.7 ± 14.4	0.076	74.4 ± 14.4	0.788	26.7 ± 14.1	0.047 *	58.9 ± 17.0	0.449	72.9 ± 13.0	0.00 **	13.9 ± 18.8	0.026 *	45.4 ± 19.2	0.869	76.3 ± 15.3	0.882	30.9 ± 17.7	0.953
None	86 (35.2)	43.8 ± 17.7		74.9 ± 15.0		31.1 ± 17.9		60.6 ± 15.3		80.0 ± 13.4		19.5 ± 18.0		44.9 ± 23.8		75.9 ± 18.5		31.0 ± 24.6	

** *p* < 0.001, * *p* < 0.05.

**Table 6 ijerph-19-06633-t006:** *T* test analysis of trainees’ score progress before and after the epidemic.

Training Year	Dimensions	*n*	Mean (T2 − T1) ^1^	SD	*t*	*df*	*p* ^2^
2019 Before the epidemic	Sizing Up and Decision Making	86	31.12	17.86	16.158	85	0.000 **
	Fire Communication	86	19.50	18.01	10.043	85	0.000 **
	Incident Safety Management	86	31.03	24.63	11.687	85	0.000 **
2020–2021 Epidemic continues	Sizing Up and Decision Making	158	26.66	14.05	23.851	157	0.000 **
	Fire Communication	158	13.94	18.78	9.333	157	0.000 **
	Incident Safety Management	158	30.86	17.69	21.927	157	0.000 **

^1^ T2: Post-Test scores, T1: Pre-Test scores; ^2^ ** *p* < 0.001.

**Table 7 ijerph-19-06633-t007:** Variance analysis of progress before and after the epidemic (ANOVA).

Dimensions (T2 − T1) ^1^	*df*	RMS	*F*	*p* ^2^
Sizing Up and Decision Making	1	1103.626	4.596	0.033 *
Fire Communication	1	1719.652	5.018	0.026 *
Incident Safety Management	1	1.750	0.004	0.948

^1^ T2: Post-Test scores, T1: Pre-Test scores; ^2^ * *p* < 0.05.

**Table 8 ijerph-19-06633-t008:** *T* Test analysis of trainees’ self-assessment scores before and during the VRST training.

Dimension	Ability Indicators	Mean ± SD		*t*	*p*	*d*
		Pre-Training *n* = 129	Post-Training *n* = 115			
Sizing Up and Decision Making	The ability to analyze the situation of the fire.	3.31 ± 0.66	3.68 ± 0.49	−4.99	0.000	0.63
	The ability to use the content of the rescue plan.	3.06 ± 0.75	3.48 ± 0.57	−4.85	0.000	0.62
	The ability to allocate resources.	3.21 ± 0.74	3.5 ± 0.64	−3.32	0.001	0.43
Fire Communication	The ability to communicate with radio and verbal commands.	3.26 ± 0.78	3.49 ± 0.6	−2.50	0.013	0.32
	The ability to allocate and track task assignment personnel.	3.15 ± 0.77	3.58 ± 0.61	−4.86	0.000	0.62
	The ability to communicate and coordinate.	3.15 ± 0.73	3.44 ± 0.69	−3.24	0.001	0.42
Incident Safety Management	The ability to respond to emergencies.	3.31 ± 0.74	3.57 ± 0.61	−3.03	0.003	0.39
	The ability to implement safety management.	3.13 ± 0.73	3.5 ± 0.63	−4.14	0.000	0.53
	The ability to transfer command.	3.08 ± 0.77	3.39 ± 0.62	−3.49	0.001	0.45

**Table 9 ijerph-19-06633-t009:** Summary of this study’s satisfaction survey results.

	Construction	No.	Questions	Satisfaction Level		
				Mean (Mode)	Top Box	Bottom Box
Learning Satisfaction	Effectiveness	05	How much does the training in the fire simulation scenario increase your commanding ability in an actual fire scene?	4.23 (4)	37.74%	0%
		04	How much do you think using VR as tools in lessons help your learning skills?	4.14 (4)	28.30%	0%
		06	How much does the VR help you familiarize with fire response guidelines / SOP?	4.25 (4)	38.68%	0%
	Professionalism rating	03	What is your rating for the entire course?	4.56 (5)	60.38%	0%
		02	What is your rating on the opinions given by instructors and evaluators?	4.71 (5)	70.75%	0%
	Confidence	08	Are you willing to participate in another VR fire rescue training?	4.24 (5)	48.11%	0%
		07	How well did you make decisions as an IC during the simulation?	4.41 (5)	49.06%	0%
		01	How much does the lesson you’ve learned help in fire command?	4.63 (5)	66.98%	0%

## Data Availability

This study’s data was used with the permission of the Chief Operating Officer of the ERCA.

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
