# Peer review of "The Study of Alternative Fire Commanders’ Training Program during the COVID-19 Pandemic Situation in New Taipei City, Taiwan"

_ijerph, 2022, doi:10.3390/ijerph19116633_

Round 1

Reviewer 1 Report

Overall, the authors made significant improvements to the paper. Some of the tables and images are hard to follow/ are not easy for the reader comprehend/understand. I also would like to see more of how this research impacts the larger community and how we use this research in the future/ outside the fire service in Taiwan. How can the fire service/police serve and more tactical populations use this information? Please make sure text size and font is consistent throughout the tables/figures in the paper. How the paper is presently written it is very specific to the fire service in Taiwan- this is a small audience and the paper could be strengthen by making it more applicable to the general population/other fire service professionals. It would also make the paper more meaningful to make a connection as to how this research helps society as whole. 

Author Response

Dear reviewer,

We should like to express my appreciation to you for suggesting how to improve our paper. We have answered all your questions item by item and updated the manuscript (please see the attachment). Also, we have checked for repeated words in the article and deleted them. The article has been properly revised and corrected by native speakers. I hope to obtain your approval for this article to be published.

Kind regards,

Manuscript ID: ijerph-1732612

Reviewer 2 Report

I have read the cover letter and the authors’ point-by-point responses as well as all the modifications made to the manuscript. I believe that the authors have made a considerable effort to adopt most of the suggestions included in my review, especially the ones that I consider most important.
The paper has been improved significantly and I am willing to accept the paper in its current form.  

Author Response

Dear reviewer,

I am glad to receive your comments on this article and that you agreed to accept it. Thank you for your contribution to the revision of this article! 
I wish you good health and happiness!
Kind regards,

Manuscript ID: ijerph-1732612
Article Title: The Study of Alternative Fire Commanders' Training Program during the COVID-19 Pandemic Situation in New Taipei City, Taiwan

Reviewer 3 Report

Accepted.

Author Response

Dear reviewer,

I am glad to receive your comments on this article and that you agreed to accept it. Thank you for your contribution to the revision of this article! 
I wish you good health and happiness!
Kind regards,

Manuscript ID: ijerph-1732612
Article Title: The Study of Alternative Fire Commanders' Training Program during the COVID-19 Pandemic Situation in New Taipei City, Taiwan

This manuscript is a resubmission of an earlier submission. The following is a list of the peer review reports and author responses from that submission.

Round 1

Reviewer 1 Report

Dear authors,

topic of your paper is very up to date and applicable. It is interesting to see that VR can be used to provide trainings during this period of pandemic situation when restrictions do not allows performing these educations normally. However the main weaknesses of your paper are scientific soundness and quality of presentation. In introductory part of paper it will be useful to present reader what is the research gap that you identified in the literature. You identified problem in the practice, but you need to better elaborate and connect it with previous scientific research done in this field.

I suggest making better structure of the paper. You do not need so many subtitles in organizing sections of your paper. I suggest following structure:

1. Introduction (Please briefly describe current state of research field, define the purpose of your study, and highlight what will be the contribution of your study)

2. Materials and Method

2.1. Experiment process

2.2. Training Setup

2.3.  Virtual Scenario Design

2.4. Instrument design (describe research instrument – the evaluation form and feedback questionnaire)

2.5. Data analysis (describe what statistical analysis you will perform and reasons for precisely applying these analysis)

3. Results

4. Discussion

5. Conclusions

You mention in the paper limitations of the training, but please provide limitation of your study. The authors need to better elaborate contribution of the study and give more future research directions.

The paper needs to be proofread by native English speaker and references need to be corrected according to the journal guidelines.

Author Response

Dear reviewer

Thank you for the reviewer’s suggestions. We should like to express my appreciation to you for suggesting how to improve our paper. I have responded point-by-point to your comments. Please see the attachment, hoping to gain your approval.

Best wishes,

Reviewer 2 Report

This paper presents a study of the method for fire commanders' training through Virtual Scenario. 244 fire commanders participated in the training and the training results were evaluated by a homemade questionnaire with three dimensions. Due to the following comments, I do not recommend acceptance of this paper for publication in International Journal of Environmental Research and Public Health.

Frist, one of the main objectives of the present study is the validation of a training method for fire commanders during the epidemic prevention period. However, the experiment was started before the outbreak of COVID-19, whether the factors of epidemic prevention period were taken into account in the setting of training scenarios? For example, the number of people in some public buildings would decrease during the epidemic prevention period.

Second, the reliability of the questionnaire question settings should be further discussed. For example, the correlation analysis results of common factors, standardized path coefficient and critical ratio of path coefficient in final modified model as well as model fit and corresponding indicators should be further discussed.

Third, just as mentioned by the authors, this study is limited to building fire. However, there are many fire scenarios in reality. The applicability of the conclusions of this study is questionable.

At the same time, the effects of high temperature and toxic smoke generated by the fire were not considered in the VR training. The absence of these parameters may make the experiment meaningless.

Author Response

Dear reviewer

Thank you very much for your suggestion. It made me realize that there are still many unclear points in my manuscript. In the revised version, I have explained point-by-point responding to your doubts. Please see the attachment, hoping to gain your approval.

Best wishes,

Reviewer 3 Report

  • abstract is a little unclear and hard to follow 
  • line 33 please define "level 3" what does this mean? People outside of Taiwan do not know what this means. 
  • line 36 change high risked to high risk 
  • line 45 might worth added virtual reality training as that is what was used in the study 
  • line 50 does the paragraph need to be included? What is this from? 
  • I am not sure paragraph 1.2.1 is needed or really makes sense to provide such detailed date by date progression of Covid-19 perhaps a more general paragraph summarizing the global devastation that Covid-19 has had and explain how lock-downs have impacted society as whole would be more appropriate
  • Paragraph 1.2.2 I would provide more detail of what traditional training entails and explain how this was not feasible due to the pandemic 
  • Perhaps I missed this but a table with the demographic information of the participants would be a sound addition, (males vs females, age, height, weight, years of service?) 
  • Is there a way to compare scores of individuals who participated in these trainings in person vs the VR format? Is there previous data that can be used? You could find age matched controls from previous courses?
  • line 159 please explain what the different VR formats are, individuals without VR training do not know what these are 
  • I think the introduction and lit review could be strengthen by providing a more detailed explanation of VR and how it has been used in the past. Some of this is described in the methods but using it in the intro could make the methods easier to read and understand 
  • line 263 please present data using the following format mean±SD
  • Was there a baseline measure taken? Then pre and post or just pre and post? Perhaps these changes were seeing from pre to post are just reflecting the firefighters learning how to respond to the testing and becoming familiar with the testing?
  • Table 4: What do you believe accounted for the differences in pre-test scores between years? Was years of experience different? was age different among years? was education level different among the groups? What about fitness levels? all of these could be variables effecting these scores? 
  • Lines 364-377 consider moving these to the discussion rather than the results, results are just the data and then discussion is where you interpret those
  • Figure 8 is hard to understand/could be explained in greater detail 
  • Lines 393-400 should be provided at the beginning of the methods a table would be a nice addition so readers can easily find this data 
  • Lines 412-415 state that the three dimensions were explained in the literature review however I think the literature can be improved by providing more detail and background into these dimensions and their role they play in sound firefighting 
  • Line 440 further how stress might impact this. What stress been linked to in previous research 
  • Another large limitation is you do not have a comparison between non VR learning vs VR learning? Was the VR truly as effective as in person traditional learning or did scores improve because the participants just became more familiar with the testing and protocols
  • This paper has good merit but needs to more clearly written and more detail needs to be provided in order for this to be considered for publication.

Author Response

Dear reviewer

I should like to express my appreciation to you for suggesting how to improve our paper. I have adjusted the contents one by one according to your suggestions and added explanations as much as possible, the manuscript has been adjusted drastically. We sincerely look forward to obtaining your approval and permission. Please see the attachment.

Best wishes,

Reviewer 4 Report

In this study, a Fire command course (C1) is used in combination with a VR simulation for training fire station captains.  To evaluate the effectiveness of the training method a pre-and post-test evaluation takes place and an after-class questionnaire is used to summarize the learning effectiveness. Interviews from the evaluators are also utilized.

I believe that this is an interesting study proving that VR simulations can be a viable solution in cases where other traditional training methods can not be applied due to various reasons such as the COVID-19 epidemic situation in this case.  VR can also incorporate data gathered from real-life scenarios.  In this paper historical fire cases were incorporated in the VR simulations. 

This paper proves that a certain training method (consisting of a course plus a VR simulation) is an effective method for fire commanders during the epidemic period since it reduces the chance of contact amongst trainees. 

Below I make some suggestions for the authors to consider in order the paper to be improved.

I believe that the aims and objectives of the study must be clearly stated in one place, (e.g. somewhere in section 2) perhaps by stating the research hypotheses, and the methods that will be used to answer these hypotheses.  Some research objectives are mentioned in the abstract and section 2 but this is not all. Later on in the paper, in section 3.2 we see that the authors also compare the post-test scores between two periods, before the epidemic and during the epidemic, to understand whether the adjustments made in response to the epidemic prevention regulations as well as the additional work demands during the epidemic affected the learning process and the training results.  I believe that it would be better for the readability of the paper if the all-research objectives are stated clearly somewhere in the text (for example in section 2).

Another thing that would prove the efficiency of the method is the use of past data if its available. The paper as it is proves that the training method which consists of a course plus a VR simulation is effective. It doesn't prove however if it is equally or more effective as the method that was used before the COVID-19 outbreak (before 2019), which did not involve VR simulations. 

Since it is not feasible to have a control group (i.e., a group following a traditional training method) then perhaps past data could be utilized. I assume that the same concepts were also taught and practiced before 2019 (when the VR simulations were incorporated into the training method), so If data exists from that period, it would be very good for this research if it was also utilized.  In this way, it can be proved whether VR simulations are equally or even more effective than traditional methods.

Since there is also experience about the training methods used before 2019, I think the authors should explain briefly how the training was implemented before 2019.

It would be also nice if the following questions were answered: Did past training involve firefighters besides Incident Commanders? Since firefighting requires teamwork and team coordination how is the training affected when other personnel is not involved (as in the VR case)?

Also, it would be good if they summarized the advantages and disadvantages of one method over the other (traditional vs VR) according to Professionals’ experience.

Points that need to be changed.

The word IC is used in the abstract but it is not explained what IC stands for. This is explained later on in the main text  (in page 2, 1.2.2.)   so it is better to also use the phrase "Incident Commanders" in the Abstract rather that IC alone.

Points that need clarification:

Page 3&4, "Fire Command Course (C1)". The authors do not mention how this course is delivered? is it taught in class? via teleconferencing? 
If it was taught in class what were the measures taken to avoid trainee contact?

Page 4, “we decided that Projection VR is the best choice, as it offers the most realistic experience when trainees are given the same tools (radio) to command the firefighter”.

It is understandable that projection VR is better than VR head-mounted displays in the COVID-19 period in order to avoid the spread of the virus but is it better in general for firefighter training? Why is it the best choice? As it is mentioned on page 14  head-mounted displays were utilized in other studies.

Some statistics regarding the sample (“Narrative statistics”) are given on page 13. Why aren’t these statistics given in the beginning together with any other information concerning the trainees?

Were the pre and post-tests the same? If not how do they differ? On page 7 it is mentioned that “The pre-test and post-test evaluation form in this experiment are with the same content”, and on page 13 it is mentioned that “ After the C1 course, a "post-test" (the fire situation assigned to each trainee is different from the pre-test) will be conducted”.  So the form filled by the evaluators is the same but the content is different? I think that all the information describing the pre and post-tests should be in one place.

Sentences to be considered for grammatical and syntactical errors

3.1.1 and 3.1.2 contain the same information.

Maybe it would be better if bullets are used for presenting the descriptive statistics for decision making, fire communication, and incident safety management.

con-firmed, page 8 (rather than confirmed).

Page 3. The following sentence is incomplete "This study combines the Fire Command Course 1 (hereinafter referred to as C1) of the NTFD".

Page 9, The following sentence needs to be checked “A total of 4 classes were trained in 2019, and a total of 86 trainees were trained;  in January 2020, when the COVID-19 epidemic began to spread”  (maybe by January…)

On page 10, it is mentioned that “the pre-test results in sizing up and decision making in 2019 and 2020 were not significant”. Is the word “significant” appropriate in this case since as I understand we are talking about test results and not statistical results? Maybe the word “satisfactory” is more appropriate.

Author Response

(The authors gave the same response as above.)

Round 2

Reviewer 2 Report

Accept.